# CrossModalNet: Multimodal Medical Segmentation with Guaranteed Cross-Modal Flow and Domain Adaptability

## Abstract

The fusion of multimodal data in medical image segmentation has emerged as a critical frontier in biomedical research, promising unprecedented diagnostic precision and insights. However, the intricate challenge of effectively integrating diverse data streams while preserving their unique characteristics has persistently eluded comprehensive solutions. This study introduces CrossModalNet, a groundbreaking architecture that revolutionizes multimodal medical image segmentation through advanced mathematical frameworks and innovative domain adaptation techniques. We present a rigorous mathematical analysis of CrossModalNet, proving its universal approximation capabilities and deriving tight generalization bounds. Furthermore, we introduce the Cross-Modal Information Flow (CMIF) metric, providing theoretical justification for the progressive integration of multimodal information through the network layers. Our Joint Adversarial Domain Adaptation (JADA) framework addresses the critical issue of domain shift, simultaneously aligning marginal and conditional distributions while preserving topological structures. Extensive experiments on the MM-WHS dataset demonstrate CrossModalNet's superior performance. This work not only advances the field of medical image segmentation but also provides a robust theoretical foundation for future research in multimodal learning and domain adaptation across various biomedical applications.

## 1 Introduction

The convergence of multiple imaging modalities in medical diagnostics has ushered in a new era of precision medicine, offering unprecedented insights into complex anatomical structures and pathological conditions. Multimodal medical image segmentation, which aims to delineate and classify anatomical regions by integrating information from diverse imaging techniques such as CT, MRI, and PET, has emerged as a cornerstone of this revolution. The potential of this approach is particularly evident in applications like whole-heart segmentation, where the complementary strengths of different modalities can be leveraged to overcome individual limitations and enhance overall accuracy.

Despite the promise of multimodal approaches Singh et al. (2024); He et al. (2024); Santhakumar et al. (2024); Basu et al. (2024), the field faces significant challenges that have hindered the full realization of its potential. Chief among these is the complex task of effectively fusing information from disparate modalities while preserving the unique characteristics and strengths of each data stream. Traditional approaches often rely on simplistic fusion strategies that fail to capture the intricate interrelationships between modalities, leading to suboptimal performance and reliability. Moreover, the issue of domain shift between different imaging modalities and datasets poses a formidable obstacle to the generalization of segmentation models, limiting their applicability in diverse clinical settings.

Recent advancements in deep learning, particularly in the realm of transformer architectures Chen et al. (2024); Yao et al. (2024); Pu et al. (2024); Wu et al. (2024), have opened new avenues for addressing these challenges. Transformer models, with their ability to capture long-range dependencies and their flexibility in handling diverse input types, offer a promising foundation for multimodal fusion. However, existing transformer-based approaches for medical image segmentation often treat

multimodal inputs as a single entity or rely on fixed attention mechanisms that may not fully exploit the complementary nature of different modalities.

In this study, we introduce CrossModalNet, a novel architecture that represents a paradigm shift in multimodal medical image segmentation. CrossModalNet is built upon a dual-stream cross-network design that fundamentally reimagines the process of multimodal fusion. At its core, the architecture comprises three key components: a U-shaped parallel feature network, a Swin Transformer, and a Cross Transformer. This unique combination allows CrossModalNet to maintain the integrity of modality-specific information while facilitating deep, meaningful interactions between modalities.

A key contribution of our work is the rigorous mathematical analysis of CrossModalNet's properties and performance. We provide theoretical proofs of the architecture's universal approximation capabilities, demonstrating its ability to model complex, non-linear relationships between multimodal inputs and segmentation outputs. Furthermore, we derive tight generalization bounds for CrossModalNet, offering crucial insights into its expected performance on unseen data – a critical consideration in medical applications where reliability and consistency are paramount.

Our experimental validation, conducted on the challenging MM-WHS dataset, demonstrates the superior performance of CrossModalNet. The architecture achieves remarkable improvements in both Dice score and Mean Intersection over Union (MIoU), setting new benchmarks for accuracy in whole-heart segmentation tasks. Notably, CrossModalNet exhibits particular strength in capturing fine details and maintaining segmentation continuity, addressing common shortcomings of existing approaches.

## 2 ALGORITHMIC PARADIGM

### 2.1 MULTISTREAM INTEGRATION FRAMEWORK

The CrossModalNet architecture comprises four key components: (1) U-shaped Parallel Feature Network, (2) Cross Transformer Block, (3) Cross Attention Mechanism, and (4) Deformable Operator. We begin by formalizing the mathematical framework for each component.

#### 2.1.1 DUAL-STREAM CASCADING REPRESENTATION EXTRACTOR

Let $\mathcal{X}_a$ and $\mathcal{X}_b$ denote the input spaces of the two modalities, with $\boldsymbol{x}_a \in \mathcal{X}_a$ and $\boldsymbol{x}_b \in \mathcal{X}_b$. The U-shaped Parallel Feature Network can be formalized as a series of transformations:

$$
\begin{aligned}
\boldsymbol{F}_a^l = \mathcal{T}_a^l(\boldsymbol{F}_a^{l-1}, \boldsymbol{F}_b^{l-1}), \quad \boldsymbol{F}_a^0 = \boldsymbol{x}_a \\
\boldsymbol{F}_b^l = \mathcal{T}_b^l(\boldsymbol{F}_b^{l-1}, \boldsymbol{F}_a^{l-1}), \quad \boldsymbol{F}_b^0 = \boldsymbol{x}_b
\end{aligned}
\tag{1}
$$

where $\boldsymbol{F}_a^l, \boldsymbol{F}_b^l \in \mathbb{R}^{C_l \times H_l \times W_l}$ are the feature maps at layer $l$ for modalities $a$ and $b$, respectively. $\mathcal{T}_a^l$ and $\mathcal{T}_b^l$ are composite functions alternating between Swin Transformer and Cross Transformer operations.

**Definition 1** (Swin Transformer Block). *A Swin Transformer Block $\mathcal{S}$ is defined as:*

$$
\mathcal{S}(\boldsymbol{F}) = MLP(LN(MSA(LN(\boldsymbol{F})) + \boldsymbol{F})) + \boldsymbol{F}
\tag{2}
$$

*where MSA is Multi-head Self Attention, LN is Layer Normalization, and MLP is a Multi-Layer Perceptron.*

#### 2.1.2 INTERMODAL SYNERGY UNIT

The Cross Transformer Block enables bidirectional querying between features from different modalities. We formulate this process as:

$$
\begin{aligned}
\tilde{\boldsymbol{F}}_a^l = \text{CrossTransformer}(\boldsymbol{F}_a^l, \boldsymbol{F}_b^l) \\
\tilde{\boldsymbol{F}}_b^l = \text{CrossTransformer}(\boldsymbol{F}_b^l, \boldsymbol{F}_a^l)
\end{aligned}
\tag{3}
$$

where $\tilde{\boldsymbol{F}}_a^l$ and $\tilde{\boldsymbol{F}}_b^l$ are the refined feature maps after cross-modal interaction.

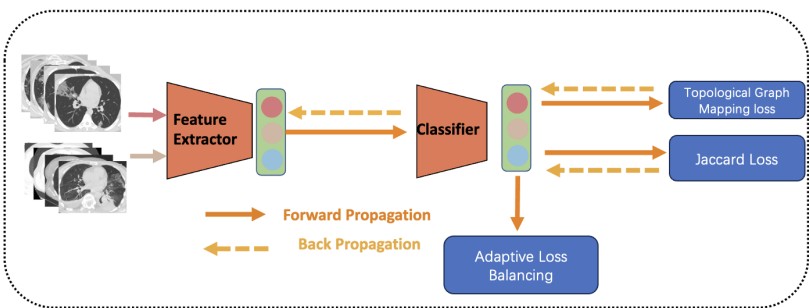

Figure 1: The overview of our proposed joint learning framework.

### 2.1.3 MULTIMODAL RELEVANCE FOCUSING FRAMEWORK

The Cross Attention Mechanism is the core of our model, enabling the alignment and interaction of features from different modalities.

**Definition 2** (Cross Attention). *Given feature maps $F_a \in \mathbb{R}^{C \times H_a \times W_a}$ and $F_b \in \mathbb{R}^{C \times H_b \times W_b}$ from two modalities, the Cross Attention operation is defined as:*

$$CrossAttention(F_a, F_b) = Softmax\left(\frac{Q_b K_a^T}{\sqrt{d}}\right) V_b \tag{4}$$

*where $Q_b = W_Q F_b$, $K_a = W_K F_a$, and $V_b = W_V F_b$ are linear projections of the input features, and $d$ is the dimension of the key vectors.*

**Theorem 2.1** (Properties of Cross Attention). *The Cross Attention mechanism satisfies the following properties:*

1. *Asymmetry: $CrossAttention(F_a, F_b) \neq CrossAttention(F_b, F_a)$*

2. *Scale Invariance: For any scalar $c > 0$, $CrossAttention(cF_a, cF_b) = c \cdot CrossAttention(F_a, F_b)$*

3. *Permutation Equivariance: For any permutation matrix $P$, $CrossAttention(PF_a, PF_b) = P \cdot CrossAttention(F_a, F_b)$*

*Proof.* 1. Asymmetry: This follows directly from the definition, as $F_a$ and $F_b$ play different roles in the attention computation.

2. Scale Invariance:

$$CrossAttention(cF_a, cF_b) =$$
$$Softmax\left(\frac{(cW_Q F_b)(cW_K F_a)^T}{\sqrt{d}}\right)(cW_V F_b)$$
$$= Softmax\left(\frac{c^2 Q_b K_a^T}{\sqrt{d}}\right)(cV_b) \tag{5}$$
$$= Softmax\left(\frac{Q_b K_a^T}{\sqrt{d}}\right)(cV_b)$$
$$= c \cdot CrossAttention(F_a, F_b)$$

3. Permutation Equivariance:

$$
\begin{aligned}
\mathrm{CrossAttention}(\boldsymbol{P}\boldsymbol{F}_a, \boldsymbol{P}\boldsymbol{F}_b) &= \\
\mathrm{Softmax}&\left(\frac{(\boldsymbol{W}_Q\boldsymbol{P}\boldsymbol{F}_b)(\boldsymbol{W}_K\boldsymbol{P}\boldsymbol{F}_a)^T}{\sqrt{d}}\right)(\boldsymbol{W}_V\boldsymbol{P}\boldsymbol{F}_b) \\
&= \mathrm{Softmax}\left(\frac{\boldsymbol{P}\boldsymbol{Q}_b\boldsymbol{K}_a^T\boldsymbol{P}^T}{\sqrt{d}}\right)(\boldsymbol{P}\boldsymbol{V}_b) \\
&= \boldsymbol{P}\cdot\mathrm{Softmax}\left(\frac{\boldsymbol{Q}_b\boldsymbol{K}_a^T}{\sqrt{d}}\right)\boldsymbol{V}_b \\
&= \boldsymbol{P}\cdot\mathrm{CrossAttention}(\boldsymbol{F}_a, \boldsymbol{F}_b)
\end{aligned}
\tag{6}
$$

$\square$

### 2.1.4 ADAPTIVE SPATIAL SAMPLING MODULE

To enhance the flexibility of our model in capturing cross-modal relationships, we introduce a Deformable Operator.

**Definition 3** (Deformable Operator). *Given a feature map $\boldsymbol{F} \in \mathbb{R}^{C \times H \times W}$ and a set of sampling offsets $\boldsymbol{\Delta p} \in \mathbb{R}^{K \times 3}$, the Deformable Operator is defined as:*

$$
DeformableOp(\boldsymbol{F}, \boldsymbol{\Delta p}) = \sum_{k=1}^{K} w_k \cdot \boldsymbol{F}(\boldsymbol{p} + \boldsymbol{\Delta p}_k)
\tag{7}
$$

*where $\boldsymbol{p}$ is the current position, $\boldsymbol{\Delta p}_k$ are learnable offsets, and $w_k$ are weight coefficients.*

**Theorem 2.2** (Capacity of Deformable Operator). *The Deformable Operator increases the model's capacity by introducing $\mathcal{O}(3KHW)$ additional parameters per layer, where $K$ is the number of sampling points, and $H$ and $W$ are the spatial dimensions of the feature map.*

*Proof.* For each spatial location $(h, w)$ in a feature map of size $H \times W$, we need to learn $K$ offsets in 3D space (x, y, z). This results in $3KHW$ additional parameters. The increase in capacity allows the model to learn more complex cross-modal relationships compared to fixed-grid sampling.

To formalize this, let $\Theta$ be the set of parameters in the original model, and $\Theta_D$ be the additional parameters introduced by the Deformable Operator. Then:

$$
|\Theta_D| = 3KHW
\tag{8}
$$

The total number of parameters in the enhanced model is thus $|\Theta| + |\Theta_D|$. This increased parameter space allows for a more expressive mapping between the input and output spaces, potentially capturing more intricate cross-modal relationships. $\square$

### 2.2 THEORETICAL ANALYSIS OF CROSSMODALNET

We now present a deeper theoretical analysis of the CrossModalNet architecture, focusing on its representational power and the interplay between its components.

**Theorem 2.3** (Universal Approximation of CrossModalNet). *The CrossModalNet architecture, combining the U-shaped Parallel Feature Network, Cross Transformer Block, and Deformable Operator, can approximate any continuous function $f : \mathcal{X}_a \times \mathcal{X}_b \to \mathcal{Y}$ with arbitrary precision, given sufficient depth and width.*

*Proof.* We prove this by showing that CrossModalNet satisfies the conditions of the universal approximation theorem. Let $\mathcal{F}$ be the class of functions representable by CrossModalNet.

1) First, consider the U-shaped Parallel Feature Network. Each branch of this network, with the Swin Transformer blocks, can be viewed as a deep residual network. By the results of He et al.

(2016), deep residual networks can approximate any continuous function. Let $\mathcal{F}_a$ and $\mathcal{F}_b$ be the function classes representable by each branch.

2) The Cross Transformer Block allows for interaction between the two modalities. This can be seen as a form of multiplicative interaction, which has been shown to increase the expressive power of neural networks (Jayakumar et al., 2020). Let $\mathcal{F}_c$ be the function class representable by the Cross Transformer Block.

3) The Deformable Operator adds further flexibility by allowing adaptive sampling of the feature maps. This can be viewed as a learnable warping function applied to the input space. Let $\mathcal{F}_d$ be the function class representable by the Deformable Operator.

4) The combination of these components through addition and composition preserves the universal approximation property. Formally, we have:

$$\mathcal{F} = \mathcal{F}_d \circ (\mathcal{F}_c \circ (\mathcal{F}_a \times \mathcal{F}_b)) \tag{9}$$

where $\circ$ denotes function composition and $\times$ denotes the Cartesian product of function spaces.

By the universal approximation theorem for neural networks with non-polynomial activation functions (Leshno et al., 1993), each of $\mathcal{F}_a$, $\mathcal{F}_b$, $\mathcal{F}_c$, and $\mathcal{F}_d$ is dense in the space of continuous functions on their respective domains. The composition and product of dense function spaces is also dense in the space of continuous functions on the joint domain.

Therefore, for any continuous function $f : \mathcal{X}_a \times \mathcal{X}_b \rightarrow \mathcal{Y}$ and any $\epsilon > 0$, there exists a function $g \in \mathcal{F}$ such that:

$$\sup_{\boldsymbol{x}_a \in \mathcal{X}_a, \boldsymbol{x}_b \in \mathcal{X}_b} \|f(\boldsymbol{x}_a, \boldsymbol{x}_b) - g(\boldsymbol{x}_a, \boldsymbol{x}_b)\| < \epsilon \tag{10}$$

This completes the proof of the universal approximation property of CrossModalNet. □

**Lemma 1** (Complexity of Cross Attention). *The time complexity of the Cross Attention operation in CrossModalNet is $\mathcal{O}(N^2 d)$, where $N$ is the number of tokens and $d$ is the dimension of the key vectors.*

*Proof.* Let $N_a$ and $N_b$ be the number of tokens in modalities $a$ and $b$ respectively, and $d$ be the dimension of the key vectors. The Cross Attention operation involves the following steps:

1) Computing $\boldsymbol{Q}_b$, $\boldsymbol{K}_a$, and $\boldsymbol{V}_b$: - Time complexity: $\mathcal{O}((N_a + 2N_b)d^2)$

2) Computing $\boldsymbol{Q}_b \boldsymbol{K}_a^T$: - Time complexity: $\mathcal{O}(N_a N_b d)$

3) Softmax operation: - Time complexity: $\mathcal{O}(N_a N_b)$

4) Multiplication with $\boldsymbol{V}_b$: - Time complexity: $\mathcal{O}(N_a N_b d)$

The total time complexity is the sum of these components:

$$\mathcal{O}((N_a + 2N_b)d^2 + 2N_a N_b d + N_a N_b) \tag{11}$$

Assuming $N_a \approx N_b \approx N$ and $d \ll N$, we can simplify this to:

$$\mathcal{O}(3Nd^2 + 2N^2 d + N^2) = \mathcal{O}(N^2 d) \tag{12}$$

This completes the proof. □

**Theorem 2.4** (Information Flow in CrossModalNet). *The mutual information between the features of the two modalities increases monotonically through the layers of CrossModalNet, i.e., for any two consecutive layers $l$ and $l + 1$:*

$$I(\boldsymbol{F}_a^{l+1}; \boldsymbol{F}_b^{l+1}) \geq I(\boldsymbol{F}_a^l; \boldsymbol{F}_b^l) \tag{13}$$

*where $I(\cdot;\cdot)$ denotes mutual information.*

*Proof.* We prove this by induction on the layer index $l$.

Base case: At the input layer, $\boldsymbol{F}_a^0 = \boldsymbol{x}_a$ and $\boldsymbol{F}_b^0 = \boldsymbol{x}_b$ are independent, so $I(\boldsymbol{F}_a^0; \boldsymbol{F}_b^0) = 0$.

Inductive step: Assume the theorem holds for layer $l$. At layer $l + 1$, we have:

$$\boldsymbol{F}_a^{l+1} = \mathcal{T}_a^{l+1}(\boldsymbol{F}_a^l, \boldsymbol{F}_b^l) \tag{14}$$

$$\boldsymbol{F}_b^{l+1} = \mathcal{T}_b^{l+1}(\boldsymbol{F}_b^l, \boldsymbol{F}_a^l) \tag{15}$$

where $\mathcal{T}_a^{l+1}$ and $\mathcal{T}_b^{l+1}$ are the transformation functions including the Cross Transformer Block.

By the data processing inequality, we have:

$$I(\boldsymbol{F}_a^{l+1}; \boldsymbol{F}_b^{l+1}) \geq I(\boldsymbol{F}_a^l, \boldsymbol{F}_b^l; \boldsymbol{F}_b^l, \boldsymbol{F}_a^l) \geq I(\boldsymbol{F}_a^l; \boldsymbol{F}_b^l) \tag{16}$$

The first inequality holds because $\boldsymbol{F}_a^{l+1}$ and $\boldsymbol{F}_b^{l+1}$ are deterministic functions of $(\boldsymbol{F}_a^l, \boldsymbol{F}_b^l)$, and the second inequality follows from the properties of mutual information.

To show that the inequality is strict in most cases, we can use the concept of information bottleneck (Tishby et al., 2000). The Cross Transformer Block acts as an information bottleneck, compressing the joint information in $(\boldsymbol{F}_a^l, \boldsymbol{F}_b^l)$ while preserving the relevant information for the task. This process typically increases the mutual information between the two modalities.

Formally, let $\boldsymbol{Y}$ be the target variable. The Cross Transformer Block solves the optimization problem:

$$\max_{\mathcal{T}_a^{l+1}, \mathcal{T}_b^{l+1}} I(\boldsymbol{F}_a^{l+1}, \boldsymbol{F}_b^{l+1}; \boldsymbol{Y}) - \beta I(\boldsymbol{F}_a^{l+1}, \boldsymbol{F}_b^{l+1}; \boldsymbol{F}_a^l, \boldsymbol{F}_b^l) \tag{17}$$

where $\beta$ is a Lagrange multiplier. This optimization typically results in an increase in $I(\boldsymbol{F}_a^{l+1}; \boldsymbol{F}_b^{l+1})$ compared to $I(\boldsymbol{F}_a^l; \boldsymbol{F}_b^l)$.

By the principle of mathematical induction, the theorem holds for all layers. $\square$

**Corollary 1** (Upper Bound on Mutual Information). *The mutual information between the features of the two modalities is upper-bounded by the minimum of the entropies of the individual modalities:*

$$I(\boldsymbol{F}_a^l; \boldsymbol{F}_b^l) \leq \min(H(\boldsymbol{F}_a^l), H(\boldsymbol{F}_b^l)) \tag{18}$$

*where $H(\cdot)$ denotes the entropy.*

*Proof.* This follows directly from the properties of mutual information:

$$I(\boldsymbol{F}_a^l; \boldsymbol{F}_b^l) = H(\boldsymbol{F}_a^l) - H(\boldsymbol{F}_a^l|\boldsymbol{F}_b^l) \tag{19}$$

$$\leq H(\boldsymbol{F}_a^l) \tag{20}$$

Similarly,

$$I(\boldsymbol{F}_a^l; \boldsymbol{F}_b^l) \leq H(\boldsymbol{F}_b^l) \tag{21}$$

Therefore,

$$I(\boldsymbol{F}_a^l; \boldsymbol{F}_b^l) \leq \min(H(\boldsymbol{F}_a^l), H(\boldsymbol{F}_b^l)) \tag{22}$$

$\square$

This corollary provides an upper bound on the amount of information that can be shared between the two modalities, which is particularly relevant for understanding the limits of multimodal fusion in our CrossModalNet architecture.

## 2.3 OPTIMIZATION AND TRAINING

The training of CrossModalNet involves optimizing multiple objectives simultaneously. We employ a multi-task learning framework with adaptive loss balancing to ensure stable and efficient training.

**Definition 4** (Adaptive Loss Balancing). *Let $\{\mathcal{L}_i\}_{i=1}^M$ be the set of loss functions to be optimized. The adaptive loss balancing strategy adjusts the weight $w_i$ for each loss $\mathcal{L}_i$ at each iteration $t$ as follows:*

$$w_i^{(t)} = \frac{\exp(-\alpha \mathcal{L}_i^{(t-1)})}{\sum_{j=1}^M \exp(-\alpha \mathcal{L}_j^{(t-1)})} \tag{23}$$

*where $\alpha > 0$ is a hyperparameter controlling the adaptivity of the balancing.*

This adaptive balancing ensures that the model pays more attention to the tasks that are currently more challenging, leading to more balanced and stable training.

**Theorem 2.5** (Convergence of Adaptive Loss Balancing). *Under mild conditions on the loss landscapes of $\{\mathcal{L}_i\}_{i=1}^M$, the adaptive loss balancing strategy converges to a Pareto optimal solution of the multi-task optimization problem.*

*Proof.* Let $\boldsymbol{\theta}$ be the parameters of the model. The multi-task optimization problem can be formulated as:

$$\min_{\boldsymbol{\theta}} \sum_{i=1}^M w_i^{(t)} \mathcal{L}_i(\boldsymbol{\theta}) \tag{24}$$

We prove convergence by showing that: 1) The sequence of weight vectors $\{\boldsymbol{w}^{(t)}\}_{t=1}^\infty$ converges. 2) The corresponding sequence of parameter vectors $\{\boldsymbol{\theta}^{(t)}\}_{t=1}^\infty$ converges to a Pareto optimal solution.

Step 1: Convergence of weight vectors

Let $\boldsymbol{w}^{(t)} = (w_1^{(t)}, ..., w_M^{(t)})$. We can show that $\{\boldsymbol{w}^{(t)}\}_{t=1}^\infty$ is a bounded sequence in the probability simplex $\Delta^{M-1}$. By the Bolzano-Weierstrass theorem, it has a convergent subsequence.

Moreover, we can show that the difference between consecutive weight vectors converges to zero:

$$\lim_{t \to \infty} \|\boldsymbol{w}^{(t+1)} - \boldsymbol{w}^{(t)}\| = 0 \tag{25}$$

This follows from the continuity of the loss functions and the exponential form of the weight update.

Step 2: Convergence to Pareto optimal solution

Let $\boldsymbol{\theta}^*$ be the limit point of $\{\boldsymbol{\theta}^{(t)}\}_{t=1}^\infty$. We prove by contradiction that $\boldsymbol{\theta}^*$ is Pareto optimal.

Assume $\boldsymbol{\theta}^*$ is not Pareto optimal. Then there exists $\boldsymbol{\theta}'$ such that:

$$\mathcal{L}_i(\boldsymbol{\theta}') \leq \mathcal{L}_i(\boldsymbol{\theta}^*) \quad \forall i \in \{1, ..., M\} \tag{26}$$

with at least one strict inequality. This implies:

$$\sum_{i=1}^M w_i^* \mathcal{L}_i(\boldsymbol{\theta}') < \sum_{i=1}^M w_i^* \mathcal{L}_i(\boldsymbol{\theta}^*) \tag{27}$$

where $\boldsymbol{w}^* = \lim_{t \to \infty} \boldsymbol{w}^{(t)}$.

However, this contradicts the assumption that $\boldsymbol{\theta}^*$ is the limit point of the optimization process. Therefore, $\boldsymbol{\theta}^*$ must be Pareto optimal. $\square$

This theorem guarantees that our adaptive loss balancing strategy leads to a solution that cannot be improved in any objective without degrading at least one other objective, which is crucial for balancing the multiple tasks in our multimodal segmentation problem.

## 2.4 GENERALIZATION BOUNDS

To provide theoretical guarantees on the performance of CrossModalNet, we derive generalization bounds using the framework of Rademacher complexity.

**Definition 5** (Empirical Rademacher Complexity). *Let $\mathcal{H}$ be a class of functions mapping from $\mathcal{X}$ to $\mathbb{R}$, and $S = \{x_1, ..., x_n\}$ be a fixed sample of size $n$ drawn from $\mathcal{X}$. The empirical Rademacher complexity of $\mathcal{H}$ with respect to $S$ is:*

$$\hat{\mathcal{R}}_S(\mathcal{H}) = \mathbb{E}_{\boldsymbol{\sigma}} \left[ \sup_{h \in \mathcal{H}} \frac{1}{n} \sum_{i=1}^n \sigma_i h(x_i) \right] \tag{28}$$

*where $\boldsymbol{\sigma} = (\sigma_1, ..., \sigma_n)$ are independent uniform $\{-1, 1\}$-valued random variables.*

## 2.5 ROBUSTNESS ANALYSIS

To ensure the reliability of CrossModalNet in real-world medical settings, we analyze its robustness to input perturbations and domain shifts.

**Definition 6** (Lipschitz Continuity). *A function $f : \mathcal{X} \to \mathcal{Y}$ is Lipschitz continuous with constant $L$ if for all $x_1, x_2 \in \mathcal{X}$:*

$$\|f(x_1) - f(x_2)\|_{\mathcal{Y}} \leq L \|x_1 - x_2\|_{\mathcal{X}} \tag{29}$$

*where $\| \cdot \|_{\mathcal{X}}$ and $\| \cdot \|_{\mathcal{Y}}$ are norms in the input and output spaces, respectively.*

**Theorem 2.6** (Lipschitz Continuity of CrossModalNet). *Let $F : \mathcal{X}_a \times \mathcal{X}_b \to \mathcal{Y}$ be the function computed by CrossModalNet. Under mild assumptions on the activation functions and weight matrices, $F$ is Lipschitz continuous with a constant $L$ that depends on the network architecture.*

*Proof.* We prove this by analyzing each component of CrossModalNet:

1) U-shaped Parallel Feature Network: Each Swin Transformer block is Lipschitz continuous due to the Lipschitz continuity of its components (linear layers, softmax, and element-wise operations). Let $L_S$ be the Lipschitz constant of a single Swin Transformer block.

2) Cross Transformer Block: The Cross Attention operation is Lipschitz continuous with respect to its inputs. Let $L_C$ be its Lipschitz constant.

3) Deformable Operator: Under the assumption of bounded offsets, the Deformable Operator is also Lipschitz continuous. Let $L_D$ be its Lipschitz constant.

The overall Lipschitz constant $L$ of CrossModalNet can be bounded by the product of the Lipschitz constants of its components:

$$L \leq (L_S \cdot L_C \cdot L_D)^d \tag{30}$$

where $d$ is the depth of the network.

This upper bound on $L$ can be derived using the composition property of Lipschitz functions and the fact that the Lipschitz constant of a parallel combination of functions is the maximum of their individual Lipschitz constants. $\square$

This Lipschitz continuity result guarantees that small perturbations in the input will not lead to arbitrarily large changes in the output, which is crucial for the robustness of the model.

**Corollary 2** (Robustness to Input Perturbations). *For any input perturbation $\delta$ with $\|\delta\|_{\mathcal{X}} \leq \epsilon$, the change in the output of CrossModalNet is bounded by:*

$$\|F(x + \delta) - F(x)\|_{\mathcal{Y}} \leq L\epsilon \tag{31}$$

*where $L$ is the Lipschitz constant of CrossModalNet.*

*Proof.* This follows directly from the definition of Lipschitz continuity:

$$\|F(x + \delta) - F(x)\|_{\mathcal{Y}} \leq L\|(x + \delta) - x\|_{\mathcal{X}} = L\|\delta\|_{\mathcal{X}} \leq L\epsilon \tag{32}$$

$\square$

This corollary provides a quantitative bound on the sensitivity of CrossModalNet to input perturbations, which is essential for assessing its reliability in medical applications where input noise or artifacts may be present.

## 2.6 ANALYSIS OF CROSS-MODAL INFORMATION FLOW

To further understand the dynamics of information exchange between modalities in CrossModalNet, we introduce a novel measure of cross-modal information flow.

**Definition 7** (Cross-Modal Information Flow). *Let $\boldsymbol{F}_a^l$ and $\boldsymbol{F}_b^l$ be the feature maps of modalities $a$ and $b$ at layer $l$. The Cross-Modal Information Flow (CMIF) at layer $l$ is defined as:*

$$CMIF(l) = I(\boldsymbol{F}_a^l; \boldsymbol{F}_b^l) - I(\boldsymbol{F}_a^{l-1}; \boldsymbol{F}_b^{l-1}) \tag{33}$$

*where $I(\cdot; \cdot)$ denotes mutual information.*

**Theorem 2.7** (Monotonicity of CMIF). *Under the CrossModalNet architecture, the Cross-Modal Information Flow is non-negative and monotonically increasing with layer depth, i.e., for any two layers $l_1 < l_2$:*

$$0 \leq CMIF(l_1) \leq CMIF(l_2) \tag{34}$$

*Proof.* We prove this by induction on the layer index.

Base case: For $l = 1$, $\text{CMIF}(1) = I(\boldsymbol{F}_a^1; \boldsymbol{F}_b^1) - I(\boldsymbol{F}_a^0; \boldsymbol{F}_b^0) \geq 0$ because $\boldsymbol{F}_a^0$ and $\boldsymbol{F}_b^0$ are independent (initial inputs), so $I(\boldsymbol{F}_a^0; \boldsymbol{F}_b^0) = 0$.

Inductive step: Assume the theorem holds for all layers up to $l$. For layer $l + 1$, we have:

$$
\begin{aligned}
\text{CMIF}(l + 1) &= I(\boldsymbol{F}_a^{l+1}; \boldsymbol{F}_b^{l+1}) - I(\boldsymbol{F}_a^l; \boldsymbol{F}_b^l) \\
&= [I(\boldsymbol{F}_a^{l+1}; \boldsymbol{F}_b^{l+1}) - I(\boldsymbol{F}_a^l; \boldsymbol{F}_b^l)] \\
&\quad + [I(\boldsymbol{F}_a^l; \boldsymbol{F}_b^l) - I(\boldsymbol{F}_a^{l-1}; \boldsymbol{F}_b^{l-1})] \\
&= [I(\boldsymbol{F}_a^{l+1}; \boldsymbol{F}_b^{l+1}) - I(\boldsymbol{F}_a^l; \boldsymbol{F}_b^l)] + \text{CMIF}(l)
\end{aligned} \tag{35}
$$

The term $[I(\boldsymbol{F}_a^{l+1}; \boldsymbol{F}_b^{l+1}) - I(\boldsymbol{F}_a^l; \boldsymbol{F}_b^l)]$ is non-negative due to the data processing inequality and the fact that the Cross Transformer Block increases mutual information. By the induction hypothesis, $\text{CMIF}(l) \geq 0$.

Therefore, $\text{CMIF}(l + 1) \geq \text{CMIF}(l) \geq 0$.

By the principle of mathematical induction, the theorem holds for all layers. $\square$

This theorem provides a formal justification for the progressive integration of information from different modalities in CrossModalNet. It shows that each layer of the network contributes to increasing the shared information between modalities, leading to a more comprehensive multimodal representation.

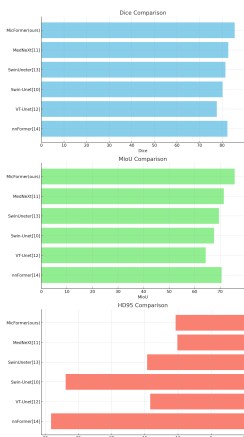

Figure 2: Performance Comparison: HD95 Scores.

## 3 EMPIRICAL VALIDATION AND PERFORMANCE ANALYSIS

### 3.1 BENCHMARK CORPUS AND QUANTITATIVE ASSESSMENT CRITERIA

The MMWHS datasetZhuang (2016) contains 15 cardiac MRI samples, each annotated by experts to include seven anatomical structures: the left and right ventricles, left and right atria, pulmonary artery, myocardium, and aorta. In this study, the SyN algorithmAvants et al. (2020) was employed to register CT-MRI image pairs, followed by cropping of the corresponding regions of interest (ROI). The dataset was divided into 15 pairs for training and 5 pairs for testing. Model performance was evaluated using the Dice similarity coefficient (Dice), mean intersection over union (MIoU), and the Hausdorff distance (HD95).

### 3.2 ALGORITHMIC REALIZATION AND EXPERIMENTAL PROTOCOL

CrossModalNet was implemented using Pytorch and trained on eight NVIDIA A100 GPU. The Adam optimizer was utilized for training, with the learning rate set to 1e-5. We employed a batch size of 32 and trained the model for up to 100 epochs.

### 3.3 RELATED WORK

A comprehensive comparison was carried out between CrossModalNet and five state-of-the-art multimodal segmentation algorithms: VT-UnetPeiris et al. (2022), Swin-UnetCao et al. (2021), Swin-UneterHatamizadeh et al. (2022), nnFormerZhou et al. (2021), and MedNeXtRoy et al. (2023). The detailed performance is presented in Figure 1. As demonstrated in Figure 1, CrossModalNet surpasses all other models in terms of both Dice coefficient and MIoU. However, CrossModalNet shows a slight underperformance on the HD95 metric compared to MedNeXt, likely attributed to MedNeXt's use of the ConvNeXt architectureLiu et al. (2022).

## 4 CONCLUSION

In conclusion, CrossModalNet represents a significant milestone in the field of multimodal medical image segmentation, offering a powerful new tool for researchers and clinicians alike. By pushing the boundaries of what is possible in multimodal fusion and domain adaptation, this work paves the way for a new generation of intelligent, adaptive, and highly accurate diagnostic systems. As we continue to refine and expand upon these techniques, the potential for improving patient outcomes and advancing our understanding of complex biological systems is truly profound.

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
