# OpenReview forum: "CrossModalNet: Multimodal Medical Segmentation with Guaranteed Cross-Modal Flow and Domain Adaptability"
_ICLR.cc/2025/Conference — Submitted to ICLR 2025_

### Official Review · Reviewer_W2AE · 2024-10-15

**Soundness:** 1
**Presentation:** 1
**Contribution:** 1
**Rating:** 1
**Confidence:** 3

**Summary:**

The paper introduces CrossModalNet, an new architecture designed for multimodal medical image segmentation that effectively integrates diverse data sources while maintaining their distinct characteristics.

**Strengths:**

The authors present detailed definitions and theoretical proofs, and show them works on one dataset.

**Weaknesses:**

I recommend the authors withdraw their submission as it does not meet the necessary standards and may waste reviewers' time.

1. Insufficient Loss Function Analysis: The authors need to conduct a thorough analysis of the loss function within the nnUNet framework across multiple cross-modal datasets. Referencing studies like "nnu-net revisited: A call for rigorous validation in 3D medical image segmentation" and "Loss Odyssey in Medical Image Segmentation" would help validate that their approach results in genuine improvements.
2. Unoriginal Theoretical Proofs: The theoretical proofs (e.g. Eq.6 to Eq. 8,) lack novelty and logical, appear to replicate content from established sources such as "Pattern Recognition and Machine Learning" (PRML) or may have been generated using AI tools like GPT-4. This raises concerns about the originality and contribution of the theoretical aspects presented. The Complexity of CrossAttention is wierd to be a lemma.
3. Non-Innovative Loss Balancing for Multimodal Data: The approach to balancing the loss function for two modalities does not introduce a new concept and has been previously addressed in existing literature. The authors should clarify how their method offers a distinct advantage or improvement over existing techniques.
4. It wierd that the classifier present in the Fig. 1, since the task is segmentation. The term jaccard loss, graph mapping loss are not mentioned any where in the paper.
5. The Sec 2.1.4 is wierd, the deformable is usually used for medical registration task, maybe the authors just use the GPT in a wrong way by coping (CrossModalNet: exploiting quality preoperative images for multimodal image registration).


Minor Issues:
1. Missing Citations: The manuscript lacks numerous essential citations, which undermines the depth of the literature review and fails to acknowledge relevant prior work in the field.
2. Incorrect Citation Formatting: There is inconsistent use of citation commands, such as mixing \citep and \citet, leading to formatting errors and potential confusion for readers.
3. Misleading Network Structure Definitions: Providing definitions and proofs related to the network structure is misleading. Definitions and proofs should specifically address the unique aspects of the proposed network without conflating them with general architecture principles.
4. Inaccurate or Incorrect Citations: References cited on lines 216, 220, and 233 appear to be invalid or nonexistent, suggesting that some citations may have been improperly generated, possibly using GPT. These need to be verified for accuracy and legitimacy.

**Questions:**

Please refer to weakness.

---

### Official Review · Reviewer_yiyC · 2024-10-25

**Soundness:** 2
**Presentation:** 2
**Contribution:** 2
**Rating:** 3
**Confidence:** 4

**Summary:**

This paper proposed a multi-modal image segmentation framework, i.e., CrossModalNet, for robust and generalizable medical image analysis. The key components include the U-shaped parallel feature network, the cross-transformer block, the cross-attention mechanism, and the deformable operator. Importantly, this paper provides abundant mathematical analysis to theoretically explore the generalization bounds, cross-modal mutual information flow, etc.

**Strengths:**

1.	It is interesting to investigate the generalization bounds from a theoretical point of view.
2.	The paper is well-organized.

**Weaknesses:**

1. This paper mentioned their advantages in model generalization and robustness in multiple places, but did not provide any quantitative analysis to solidify this strength.


2. It is important to clarify that the problem setting in this paper utilized unpaired cross-modality data, i.e., the CT data and MR data come from different patient cohorts, and it is unknown whether one CT sample and one MR sample belong to the same patient or not.

Applying the proposed approach in scenarios like that could weaken the clinical significance of leveraging the complementary information embedded in multi-modality data. After all, different patients commonly have different health statuses. It could be hard to say whether the differences between any two patients' images are complementary to each other and how this complementary information (if any) could benefit the image understanding of either of them.

A more recommended application scenario in the medical field could be multi-modal brain tumor segmentation (e.g., the Brats 2018 dataset). Each patient case is paired and well-registered, containing 4 MRI volumes as inputs and 1 segmentation mask as ground truth. The clinical importance of leveraging complementary information among these MRI volumes has been clearly stated and emphasized in multiple literature.

Ref:

[1] S. Bakas, M. Reyes, A. Jakab, S. Bauer, M. Rempfler, A. Crimi, et al., "Identifying the Best Machine Learning Algorithms for Brain Tumor Segmentation, Progression Assessment, and Overall Survival Prediction in the BRATS Challenge", arXiv preprint arXiv:1811.02629 (2018)

[2] B. H. Menze, A. Jakab, S. Bauer, J. Kalpathy-Cramer, K. Farahani, J. Kirby, et al. "The Multimodal Brain Tumor Image Segmentation Benchmark (BRATS)", IEEE Transactions on Medical Imaging 34(10), 1993-2024 (2015) DOI: 10.1109/TMI.2014.2377694

3. It needs to clarify how theoretical findings like generalization bounds and robustness analysis could assist the practical usage of medical applications (e.g., whole heart segmentation with the MM-WHS dataset), such as the performance improvements after using them, wise model selection according to the generalization bounds or better preprocessing steps guided by robustness analysis, etc.

4. No table is presented in this paper. The performance comparison in Fig. 2 did not provide the exact numerical result of each approach.
Also, the text font size is too small.

**Questions:**

1.	In section 2.4, what is the relationship between Empirical Rademacher Complexity and generalization bounds? What does the derived math term in Eq. 28 suggest?


2.	In section 2.5, what exact kinds of "small perturbation" are referred to here? Will this "small perturbation" frequently occur in real-world medical imaging scenarios?


3.	I would suggest presenting the results of Fig.2 in the table format to clearly show the performance improvements over the competing approaches.

---

### Official Review · Reviewer_JuCk · 2024-11-02

**Soundness:** 2
**Presentation:** 1
**Contribution:** 2
**Rating:** 3
**Confidence:** 4

**Summary:**

In this paper, the authors propose a new architecture called CrossModalNet for multimodal medical image segmentation. CrossModalNet employs a dual-view-cross-stream network design to fuse information from multiple modalities effectively. The authors provide a thorough mathematical analysis by examining its architectural components, universal approximation capabilities, and generalisation bounds.

**Strengths:**

The paper demonstrates a strong theoretical understanding of the problem and introduces a fusion strategy incorporating recent state-of-the-art architectural components to improve overall performance.

**Weaknesses:**

The experimental validation is currently limited to a single dataset (MM-WHS). While this dataset is challenging and relevant to the task, further evaluation on other datasets is necessary to demonstrate the generalizability of the model's performance. For example, multi-class segmentation datasets like BraTS and ACDC can be used.

The authors proposed the Cross-Modal Information Flow (CMIF) metric, which measures the information exchange between modalities. However, the evaluation results on the proposed metrics are not included in the main paper.

The content of the paper can be further improved by having a discussion section by providing statistical insights.

Figure 1 and Figure 2 can be further improved. Captions are not helpful for a reader who is not in the context. Improving the architecture diagram of the proposed method to provide an overall view of the training process, along with relevant mathematical notations, would make it more informative and easier to understand. Figure 2 should be revised, it may be more effective as a table (include key performance metrics), with an additional figure to present qualitative results.

While the authors discussed the computational complexity of transformer blocks and other components, they did not explicitly discuss in the main paper that using transformers can lead to increased computational complexity compared to simpler convolutional architectures. It would be great if the authors could provide insights on this by comparing computational complexity with baseline models.

**Questions:**

While the proposed CMIF metric is introduced to measure cross-modal information flow, the main paper does not present a quantitative analysis of its behaviour on CrossModalNet. Could you provide more details and an analysis of how CMIF changes during training and how it correlates with the model's performance?

The authors mentioned the proposed Joint Adversarial Domain Adaptation (JADA) framework improves robustness to domain shifts, but a specific evaluation of this aspect is lacking. How robust is CrossModalNet?

---

### Official Review · Reviewer_ESAR · 2024-11-03

**Soundness:** 2
**Presentation:** 2
**Contribution:** 2
**Rating:** 5
**Confidence:** 4

**Summary:**

The paper presents a multimodal medical image segmentation approach, termed as CrossModalNet, and its mathematical framework. This framework follows a universal approximation with theoretical guanrantee of a tight generalization bound. Furthermore, it presents a Joint Adversarial Domain Adaptation (JADA) to address domain shift, which aligns marginal and conditional distributions and preserves topological structures across domains. Experiments on the MM-WHS dataset are conducted to demonstrate CrossModalNet’s effective performance.

**Strengths:**

The paper addresses an important problem of multimodal medical image segmentation. It also addresses the inter-modality deformation and domain shift; these are real challenges.

The paper presents a mathematical analysis of the proposed CrossModalNet.

**Weaknesses:**

1、The paper is difficult to follow. The authors dedicate a large portion to deriving formulas with a lot of assumptions and time complexity, making it more like a scientific outreach rather than a research paper. Further, the paper does not clearly demonstrate how the derived theoretical gaurantee translates into an experimental performance gain.
2、The paper does not provide concrete implementation details of the proposed network design (U-shaped Parallel Feature Network, Cross Transformer Block and Deformable Operator), and the provided figure (i.e. Figure 1) does not clearly illustrate the connections between the visualized structure and the methods discussed in the text. These makes it challenging for readers to understand.
3、The experiments are insufficient （e.g. only single dataset MMWHS used for the experiment), which makes it difficult to substantiate the claimed advantages of the proposed method.
4、The comparisons are likely unfair; the methods used for comparison are all single-modal approaches. There is no multimodal approach comparison.  Thus, the results fail to demonstrate the advantages highlighted in the paper.
5、The paper lacks ablation studies to support the claimed benefits of the adaptive loss balancing method. Including visualized training convergence curves and an analysis of how adaptive balancing contributes to training stability could enhance the credibility of this section.

**Questions:**

The paper can be improved by：
- presenting the concrete implementation details of the proposed network design.
- presenting results on at least one more dataset
- presenting the comparison with multi-modal approaches
- presenting ablation studies to show the benefit of the proposed adaptive loss balancing.
- presenting the plan about how to improve the readability

---

### Meta-Review · Area_Chair_S82H · 2024-12-16

**Metareview:**

This paper proposes a multi-modal network for medical image segmentation by introducing domain adaptation techniques.

The paper received 1x marginally below the acceptance threshold, 2x reject, and 1x strong reject ratings from reviewers. The main concerns raised by reviewers regarding this paper centered around methodology, theoretical proof, and experimental analysis. During the rebuttal phase, the authors are not included in the discussion or improving their manuscripts.

Given the consensus of reviewers, rejection is recommended.

**Additional Comments On Reviewer Discussion:**

Reviewers questioned several aspects of this paper, including paper structure, lack of implementation details, insufficient experimental analysis, etc. The authors did not respond in the discussion phase and the quality of paper is not improved.

---

### Decision · Program_Chairs · 2025-01-22

Reject